

# Controls on the composition of hydroxylated isoGDGTs in cultivated ammonia oxidizing Thaumarchaeota

Devika Varma[1], Laura Villanueva[1,2], Nicole J. Bale[1], Pierre Offre[1], Gert-Jan Reichart[2,3], Stefan Schouten[1,2]

[1]Department of Marine Microbiology and Biogeochemistry, NIOZ Royal Netherlands Institute for Sea Research, Den Burg, the Netherlands
[2]Department of Earth Sciences, Utrecht University, Utrecht, the Netherlands
[3]Department of Ocean Systems, NIOZ Royal Netherlands Institute for Sea Research, Den Burg, the Netherlands

*Correspondence to*: Devika Varma (devika.varma@nioz.nl)

**Abstract.** Membrane lipids of ammonia-oxidizing Thaumarchaeota, in particular isoprenoidal glycerol dialkyl glycerol tetraethers (isoGDGTs) and hydroxylated isoGDGTs (OH-isoGDGTs), have been used as biomarkers and as proxies in various environments. Controlled growth experiments have been used to investigate the factors that influence the composition of these lipids, in particular on how these factors affect the $TEX_{86}$ temperature proxy, which is based on the degree of cyclization of

isoGDGTs. Recently, the ring index of OH-isoGDGTs (RI-OH'), based on cyclization patterns of OH-isoGDGTs, and the relative abundance of OH-isoGDGTs (%OH) have emerged as promising temperature proxies. Here, we examined the impact of growth temperature and growth phase on the distribution of OH-isoGDGTs and their associated proxies using cultures of two thaumarchaeotal strains. Analysis of core lipids and headgroup compositions of isoGDGTs and OH-isoGDGTs showed no consistent differences between the mid-exponential and stationary phases for both strains. *Nitrosopumilus adriaticus* NF5

shows a substantially higher relative abundance of OH-isoGDGTs (~49 %) compared to *Nitrosopumilus piranensis* D3C (~5 %) and also relative to observations reported for core lipids in the marine environment (< 17 %), indicating large variations in %OH values even among closely related species. Unlike in the marine environment, the %OH did not decrease with increasing temperatures in either of the strains, possibly reflecting a threshold below 15 °C for this response in natural environment. The RI-OH' increases with increasing temperature in cultures of both strains, similar to the ring index of regular isoGDGTs. The

relative abundances of the headgroups varied between strains and did not respond to changes in temperature nor growth phase. The %OH and RI-OH' calculated from intact polar lipids with different headgroups revealed large differences between the distinct intact polar lipids, similar to that previously observed for regular isoGDGTs. Together, our findings suggest that growth temperature has a pronounced effect on the degree of cyclization in isoGDGTs and OH-isoGDGTs, in contrast to the relative abundance of OH-isoGDGTs, which mainly exhibits interspecies variability.



## 1 Introduction


Isoprenoidal GDGTs with 0 to 4 cyclopentane moieties, as well as crenarchaeol and its isomer (cren′), which contain a cyclohexane moiety in addition to four cyclopentane moieties are the main membrane lipids synthesized by ammonia oxidizing archaea of Thaumarchaeota (Sinninghe Damsté et al., 2002; Zeng et al., 2019), now referred to as *Nitrososphaeria* (Genome Taxonomy Database; Rinke et al., 2021). They are widely used as biomarkers and in various environmental proxies (e.g. Blaga

et al., 2009; Weijers et al., 2011; Zhang et al., 2011; Sinninghe Damsté et al., 2012; Taylor et al., 2013; Inglis et al., 2015; O'Brien et al., 2017). These include the Tetraether index of 86 carbons (TEX$_{86}$; Schouten et al., 2002) and the Ring index of isoGDGTs based on the degree of cyclization of the isoGDGTs (RI$_{isoGDGTs}$; Pearson et al., 2004; Zhang et al., 2016), which are indicative of temperature conditions at which these organisms grow.

Hydroxylated isoGDGTs (OH-isoGDGTs) are lipids with one or more hydroxy groups in the biphytanyl chains and are also

widespread in the marine environment (Liu et al., 2012b; Huguet et al., 2013; Varma et al., 2024). OH-isoGDGTs with 0 to 2 cyclopentane moieties are typically ubiquitously found in the open ocean, whereas those with 3 and 4 cyclopentane moieties, as well as those with a cyclohexane moiety and four cyclopentane moieties (OH-crenarchaeol) are mainly found in cold seep sediments, Black Sea water column and thaumarchaeotal cultures (Zhang et al., 2023; Elling et al., 2017). OH-isoGDGTs with 0 to 2 cyclopentane moieties are thought to be produced primarily by Thaumarchaeota in the marine environment (Elling et

al., 2017; Sinninghe Damsté et al., 2012; Liu et al., 2012b, a) and are utilized in paleothermometers such as %OH (Huguet et al., 2013), RI-OH', RI-OH (Lü et al., 2015), and recently TEX$_{86}^{OH}$ (Varma et al., 2024). Although OH-isoGDGT-based proxies have demonstrated promise in paleoenvironmental reconstructions (e.g. Davtian et al., 2019; Fietz et al., 2020; Morcillo-Montalbá et al., 2021; Liu et al., 2022; Sinninghe Damste et al., 2022; Davtian and Bard, 2023; Varma et al., 2024), the physiological role of OH-isoGDGTs is not well understood, which limits the interpretation of the OH-isoGDGT-based proxies.

Biophysical modelling suggests that hydroxylation enhances the fluidity of the membrane and thereby provides a mechanism for archaeal cells to adapt to lower temperatures (Huguet et al., 2017).

The isoGDGT and OH-isoGDGT-based temperature proxies are mostly established using core-top and suspended particulate matter studies (e.g. Kim et al., 2010; Basse et al., 2014a; Xie et al., 2014; Lü et al., 2015, 2019; Hurley et al., 2018; Fietz et al., 2020; Varma et al., 2024). While mesocosm studies provided insights into the impact of temperature and salinity on

isoGDGT lipid distribution and archaeal communities (Schouten et al., 2007; Wuchter et al., 2004), studies of cultivated archaea in laboratory settings allowed studying the impact of specific parameters on lipid compositions of these archaeal cultures. For example, it was shown that the degree of cyclization, as indicated by RI$_{isoGDGTs}$, increases with higher growth temperatures in cultivated Thaumarchaeota (Qin et al., 2015; Elling et al., 2015). However, the relationship between TEX$_{86}$ and temperature in these cultures is complex and differs from that observed in the natural environment (Qin et al., 2015; Elling

et al., 2015, 2017; Bale et al., 2019). Both RI$_{isoGDGTs}$ and TEX$_{86}$ exhibit distinct species-specific relationships with growth temperature among different strains of ammonia oxidizing archaea (Elling et al., 2017, 2015; Bale et al., 2019; Qin et al., 2015; Pitcher et al., 2011; Sinninghe Damsté et al., 2012). Therefore, besides environmental and ecological factors, variations in



archaeal community composition across biogeographic regions likely also play a role in shaping the relationship between isoGDGT distribution and temperature (e.g. Trommer et al., 2009; Polik et al., 2018; Besseling et al., 2019; Sollai et al., 2019).

Availability of oxygen, rate of ammonia oxidation and the culture growth phase are also known to impact the composition of the isoGDGTs in archaeal cells and thereby $TEX_{86}$ values in culture experiments (Qin et al., 2015; Hurley et al., 2016; Elling et al., 2014, 2015). However, pH and salinity have only a minor effect on $TEX_{86}$ (Elling et al., 2015; Wuchter et al., 2004). The polar headgroups attached to the core GDGTs vary among different archaeal species, core isoGDGT types, growth phases and temperatures (Bale et al., 2019; Elling et al., 2014, 2015, 2017), suggesting complex archaeal membrane adaptation.

In contrast to regular isoGDGTs, OH-isoGDGTs have been studied less extensively in cultured organisms, but the few studies do show major differences between species. Among cultured Thaumarchaeota, OH-isoGDGTs were mainly detected within the order *Nitrosopumilales* (previously reported as Group I.1a Thaumarchaeota and SAGMCG-1; cf. Elling et al., 2017) from soil and marine environments, and they were also detected in one culture of *Nitrososphaerales* (Bale et al., 2019). However, OH-isoGDGTs are not detected in all species of *Nitrosopumilales*, for instance, in the thermophilic Thaumarchaeota

*Nitrosotenuis uzonensis* (Bale et al., 2019). Similar to regular isoGDGTs, OH-isoGDGTs also exhibit variations in IPL composition across thaumarchaeotal strains (Elling et al., 2017, 2015). While OH-isoGDGT core lipids are utilized as temperature proxies in the marine environment (Huguet et al., 2013), the abundance of OH-isoGDGTs from both core lipids and IPLs in cultures does not seem to be related to the cultivation temperature of these archaea (Elling et al., 2017, 2015). Nevertheless, only a small number of archaea have yet been examined for OH-isoGDGTs, and with the development of new

proxies based on OH-isoGDGTs (cf. Varma et al., 2024), there is a need for controlled growth studies on the role of OH-isoGDGTs in membranes of archaea.

To further investigate the factors controlling the relative abundance and composition of OH-isoGDGTs, we examined the intact polar lipids of isoGDGTs and OH-isoGDGTs in two strains of Thaumarchaeota grown at different temperatures and harvested at different growth phases. We aim to throw light on the impact of temperature, growth phase and species on the

composition of OH-isoGDGTs and the proxies based on them.

## 2 Materials and Methods

### 2.1 Cultivation of AOA

Enrichment cultures of *Nitrosopumilus piranensis* D3C strain and *Nitrosopumilus adriaticus* NF5 strain were grown in 2 L Schott bottles in Synthetic Crenarchaeota Medium, supplemented with bovine liver catalase (Sigma-Aldrich, Germany) at 5 U

ml$^{-1}$ final concentration (Martens-Habbena et al., 2009; Bayer et al., 2019a, b). The growth phase of the cultures was monitored by measuring nitrite production using Griess reagent (Griess, 1879) prepared with 1 % (w/v) sulfanilamide and 0.1 % (w/v) N-naphtylethylendiamindihydrochloride in 5 % hydrochloric acid, and absorbance spectroscopy (Bayer et al., 2016) using a spectrophotometer (Molecular Devices SpectraMax M2) at absorbance 545 nm. Samples for cell counts (0.5 ml) were fixed with glutaraldehyde (0.5 % w/v) for 15 min, flash frozen with liquid nitrogen and subsequently stored at -80°C until further



use. Subsequently, the samples were diluted in TE buffer and stained with SYBR Green I (final concentration $5 \times 10^{-4}$ of the commercial stock; Life Technologies, Netherlands) prior to flow cytometry analysis to determine cell abundances by using a BD FACSCelesta flow cytometer.

## 2.2 Temperature experiment set-ups

The *N. adriaticus* NF5 strain was grown at 20, 25 and 30 °C and the *N. piranensis* D3C strain was grown at 25, 30 and 35 °C,
based on the optimal growth temperatures for each strain at a volume of 1.6 L in 2 L Schott bottles without shaking. Efforts to cultivate the strains at lower temperatures were unsuccessful, potentially due to extremely slow growth rates. The cultures were harvested at mid-exponential phase and late exponential/ stationary phase as determined by the nitrite concentration of the cultures for each temperature experiment using glass fiber filters of 0.3 µm diameter pore size (GF-75, Advantec, Japan) and stored at -80 °C until further use.

## 105    2.3 Lipid extraction and analysis

The GF-75 filters were freeze-dried and extracted using a modified Bligh and Dyer method (Bligh and Dyer, 1959; Sturt et al., 2004; Schouten et al., 2008). The filters were extracted twice ultrasonically in a mixture of methanol (MeOH), dichloromethane (DCM), and phosphate buffer (2:1:0.8, v:v:v) for 10 min. The supernatant was collected after centrifuging the mixture at 3000 rpm for 2 min. To separate the phases, additional DCM and phosphate buffer were added to the combined supernatants, resulting in a final solvent ratio of 1:1:0.9 (v:v:v). The organic phase was collected after centrifuging the mixture at 3000 rpm
for 2 min, and then the aqueous phase was extracted thrice using DCM. These steps were repeated with the same residue but using a mixture of MeOH, DCM, and trichloroacetic acid (2:1:0.8, v:v:v). The final combined extract was then dried under $N_2$ and stored at -20 °C until analysis. Prior to analysis, an internal standard, deuterated diacylglyceryltrimethylhomoserine (DGTS D-9, Avanti Polar Lipids) was added to the extracts. The extracts with the internal standard were then redissolved in
MeOH:DCM (9:1, v:v) and filtered through 0.45 µm, 4 mm diameter regenerated cellulose syringe filter (BGB analytik, USA). IPLs were analyzed according to Wörmer et al. (2013) with modifications as described in Bale et al. (2021). Briefly, samples were analyzed using Agilent 1290 Infinity I UHPLC (ultra-high-performance liquid chromatography) coupled to a Q Exactive Orbitrap HRMS (high-resolution mass spectrometry) with Ion Max source and heated electrospray ionization (HESI) probe. The separation was achieved on Acquity BEH C18 column (Waters, 2.1×150 mm, 1.7 µm particle). A solvent gradient of
MeOH: $H_2O$: formic acid: aqueous ammonia (85:15:0.12:0.04, v:v) (solvent A) and MeOH: isopropanol: formic acid: aqueous ammonia (50:50:0.12:0.04, v:v) (solvent B) was used at a constant flow rate of 0.2 ml min$^{-1}$ starting with 95 % A for 3 min, followed by decreasing A in a linear gradient to 40 % in 12 min and to 0 % in 50 min and maintaining this until 80 min. Lipids were detected using positive ion monitoring of *m/z* 350–2000 and the 10 most abundant ions of MS$^1$ scan was used to obtain data-dependent MS$^2$ spectra. In addition, we used dynamic exclusion (6.0 s) and an inclusion list with calculated *m/z* values
for all known isoGDGTs. Fragmentation was achieved using stepped normalized collision energy of 15, 22.5, and 30. isoGDGT and OH-isoGDGTs were quantified by integration of the appropriate peaks of the summed mass chromatograms





within 3 ppm of mass accuracy of relevant molecular ions ([M+H]$^+$, [M+NH$_4$]$^+$ and [M+Na]$^+$ for isoGDGTs, and [M+H−H$_2$O]$^+$, [M+H]$^+$, [M+NH$_4$]$^+$ and [M+Na]$^+$ for OH-isoGDGTs) of IPLs which contained them as core lipids. The crenarchaeol isomer (cren′) was tentatively identified by its identical mass spectrum as that of crenarchaeol and its elution

order compared with other isoGDGTs in the reverse phase chromatography.

Our approach is different from core-top studies and most other culture studies, where core lipid compositions rather than intact polar lipid compositions are investigated. However, analysis of core lipids from archaeal biomass generally involves acid hydrolysis of lipid extracts or biomass in order to remove headgroups. Unfortunately, hydroxylated isoGDGTs, the main topic of our study, can potentially undergo partial dehydration resulting in the formation of e.g. unsaturated GDGTs (Liu et al.,

2012b). This would lead to underestimations of OH-isoGDGT abundances and biases when calculating indices of core lipids. Therefore, we restricted ourselves to analyzing the IPL composition of the archaeal biomass and inferred core lipid compositions from the IPLs.

The relative abundance of OH-isoGDGTs, (%OH ; Eq. (1); modified after adding OH-isoGDGT-3 and -4 to equation from Huguet et al., 2013), ring index of OH-isoGDGTs (RI-OH′; Eq. (2); modified after adding OH-isoGDGT with three and four

cyclopentane moieties (OH-isoGDGT-3 and -4, respectively) to equation from Lü et al., 2015),TEX$_{86}$ (Eq. (3); Schouten et al., 2002), tex86oh (Eq. (4); Varma et al., 2024) and ring index of regular isoGDGTs (RI$_{isoGDGTs}$; Eq. (5); Pearson et al., 2004) were calculated as follows:

$$\%OH = \frac{[OH\text{-}0]+[OH\text{-}1]+[OH\text{-}2]+[OH\text{-}3]+[OH\text{-}4]}{[0]+[1]+[2]+[3]+[Cren]+[Cren']+[OH\text{-}0]+[OH\text{-}1]+[OH\text{-}2]+[OH\text{-}3]+[OH\text{-}4]} \times 100 \tag{1}$$

$$RI\text{-}OH' = \frac{[OH\text{-}1]+2\times[OH\text{-}2]+3\times[OH\text{-}3]+4\times[OH\text{-}4]}{[OH\text{-}0]+[OH\text{-}1]+[OH\text{-}2]+[OH\text{-}3]+[OH\text{-}4]} \tag{2}$$

$$TEX_{86} = \frac{[2]+[3]+[Cren']}{[1]+[2]+[3]+[Cren']} \tag{3}$$

$$TEX_{86}^{OH} = \frac{[2]+[3]+[Cren']}{[1]+[2]+[3]+[Cren']+[OH\text{-}0]} \tag{4}$$

$$RI_{isoGDGTs} = 1\times\frac{[1]}{\Sigma\,GDGTs} + 2\times\frac{[2]}{\Sigma\,GDGTs} + 3\times\frac{[3]}{\Sigma\,GDGTs} + 4\times\frac{[4]}{\Sigma\,GDGTs} + 5\times\frac{[Cren]}{\Sigma\,GDGTs} + 5\times\frac{[Cren']}{\Sigma\,GDGTs} \tag{5}$$

where Σ GDGTs=[isoGDGT-0]+[isoGDGT-1]+[isoGDGT-2]+[isoGDGT-3]+[isoGDGT-4]+ [Cren]+ [Cren] , [OH − n] indicates hydroxylated isoprenoidal GDGTs with n number of cyclopentane rings and [n] indicates non-hydroxylated

isoprenoidal GDGTs with n number of cyclopentane rings.

## 3 Results & Discussion

### 3.1 Intact polar lipid composition of *N. piranensis* D3C and *N. adriaticus* NF5 strains

Two strains of ammonia-oxidizing Thaumarchaeota, *N. piranensis* D3C strain and *N. adriaticus* NF5 strain, grown at different temperatures (25, 30 and 35 °C for *N. piranensis* D3C strain, and 20, 25 and 30 °C for *N. adriaticus* NF5 strain) and harvested



at both mid-exponential and stationary phases, were analyzed for isoGDGTs and OH-isoGDGTs by performing intact polar lipid analysis. The main isoGDGT core lipids observed in the IPLs were isoGDGTs with 0 to 4 cyclopentane moieties, crenarchaeol and, tentatively, the crenarchaeol isomer (Fig. S1, Table S1). The main OH-isoGDGTs observed were those with 0 to 4 cyclopentane moieties. The headgroups of the IPLs consisted of monohexose (MH), dihexose (DH) and hexose-phosphohexose (HPH).

For *N. piranensis* D3C strain, the HPH headgroup was mainly associated with isoGDGT-0 (~18–55 %), crenarchaeol (~11–70 %), isoGDGT-1 (~5–17 %) and minor amounts of isoGDGT-2, -3 and -4 (~0.1–7 %) as core lipids (Fig. S1, Table S1). No HPH-OH-isoGDGTs were detected. Similar to HPH, IPLs with a MH headgroup consisted of isoGDGT-0 (~17–71 %), crenarchaeol (~20–44 %), isoGDGT-1 (~10–22 %), isoGDGT-2 (~3–11 %), and minor amounts of isoGDGT-3 and -4 (< 4 %) core lipids. Although the overall abundance of MH-OH-isoGDGTs was low (average ~0.03 %, considering the cultures

from different temperatures and growth phases analyzed), the OH-isoGDGT-0, -1 and -2 core lipids were detected for some of the cultures (Table S1). In contrast, IPLs with a DH headgroup had isoGDGT-1 (~1–31 %), isoGDGT-2 (~7–38 %), isoGDGT-4 (~7–36 %) and moderate amounts of isoGDGT-3 (~4–11 %) as core lipids, with lower abundances of isoGDGT-0 and crenarchaeol (< 4 %), and the crenarchaeol isomer (~1–2 %). They also contained relatively high abundance of OH-isoGDGT-1 (~1–6 %) and -2 (~2–22 %), while OH-isoGDGT-0, -3 and -4 were present in lower abundance of < 2 %. Finally,

isoGDGT-0 to -4 and crenarchaeol core lipids lacking any headgroups were also identified while OH-isoGDGT core lipids without a headgroup were not observed.

   For *N. adriaticus* NF5 strain, the IPLs with an HPH headgroup consisted mostly of isoGDGT-0 (~12–83 %), crenarchaeol (~5–31 %), isoGDGT-1 (~3–15 %), and < 4 % of isoGDGT-2, -3 and -4 as core lipids (Fig. S2, Table S2). Unlike in *N. piranensis* D3C, HPH-OH-isoGDGT-0 to -2 were detected in *N. adriaticus* NF5 strain although at relatively low abundances

of < 2 %. The MH-isoGDGTs comprised ~10–33 % isoGDGT-0, ~12–25 % crenarchaeol, ~4–8 % isoGDGT-1, and < 3 % of isoGDGT-2, -3 and -4 core lipids. MH-OH-isoGDGTs were more abundant in *N. adriaticus* NF5 strain than in *N. piranensis* D3C strain and consisted of ~18–57 % of OH-isoGDGT-0, ~6–12 % of OH-isoGDGT-1 and ~0.6–4 % of OH-isoGDGT-2. The DH headgroup was primarily associated with isoGDGT-1 (~2–20 %), isoGDGT-2 (~5–22 %), isoGDGT-3 (~3–7 %) and isoGDGT-4 (~2–9 %) core lipids among the regular isoGDGTs, with smaller proportions of < 1 % of isoGDGT-0, crenarchaeol

and crenarchaeol isomer. The OH-isoGDGT-0 to -4 were observed as core lipids for DH-OH-isoGDGTs with predominantly OH-isoGDGT-2 (~19–52 %) and OH-isoGDGT-1 (~6–47 %). Core lipids without any headgroup consisted of isoGDGT-0 to -4 and crenarchaeol, while no OH-isoGDGT core lipids were observed, similar to *N. piranensis* D3C strain.

   In order to facilitate comparisons of our IPL results with those of IPLs and core lipid results from other cultures and natural environments, we calculated the summed amount of core lipids across all headgroups and plotted the overall relative abundance

of each core lipid (Fig. 1). This assumes that all the IPLs have similar ionization efficiencies, an assumption which is likely incorrect. However, it does facilitate comparisons of trends in core lipid compositions, the main aim of our study. Furthermore, we note that there are also likely biases in reported core lipid compositions due to OH-isoGDGTs having varying ionization efficiencies compared to isoGDGTs for different laboratories (cf. Varma et al., 2024) and because OH-isoGDGTs may be




underrepresented due to losses incurred during acid hydrolysis. Due to differences in e.g. ionization efficiencies between IPL
analysis, using electron spray ionization, and core lipid analysis, using atmospheric chemical ionization, quantitative
differences are likely when comparing our results with those of core lipid composition of other cultures and environments.
However, trends, e.g. changing relative abundances with temperature, will remain comparable.

Both core lipid and headgroup composition of *N. piranensis* D3C strain and *N. adriaticus* NF5 strain show substantial
differences between the two strains (Figs. 1 and 2), especially regarding the relative abundance of OH-isoGDGTs. This
suggests that large variations in the lipid composition exist among enrichment cultures of different strains, despite that they
are derived from same environment (i.e., Adriatic surface waters) and have a relatively high genetic relatedness (Bayer et al.,
2016). Below we discuss the impact of growth phases and growth temperatures on the lipid composition of the cultures as well
as species differences.

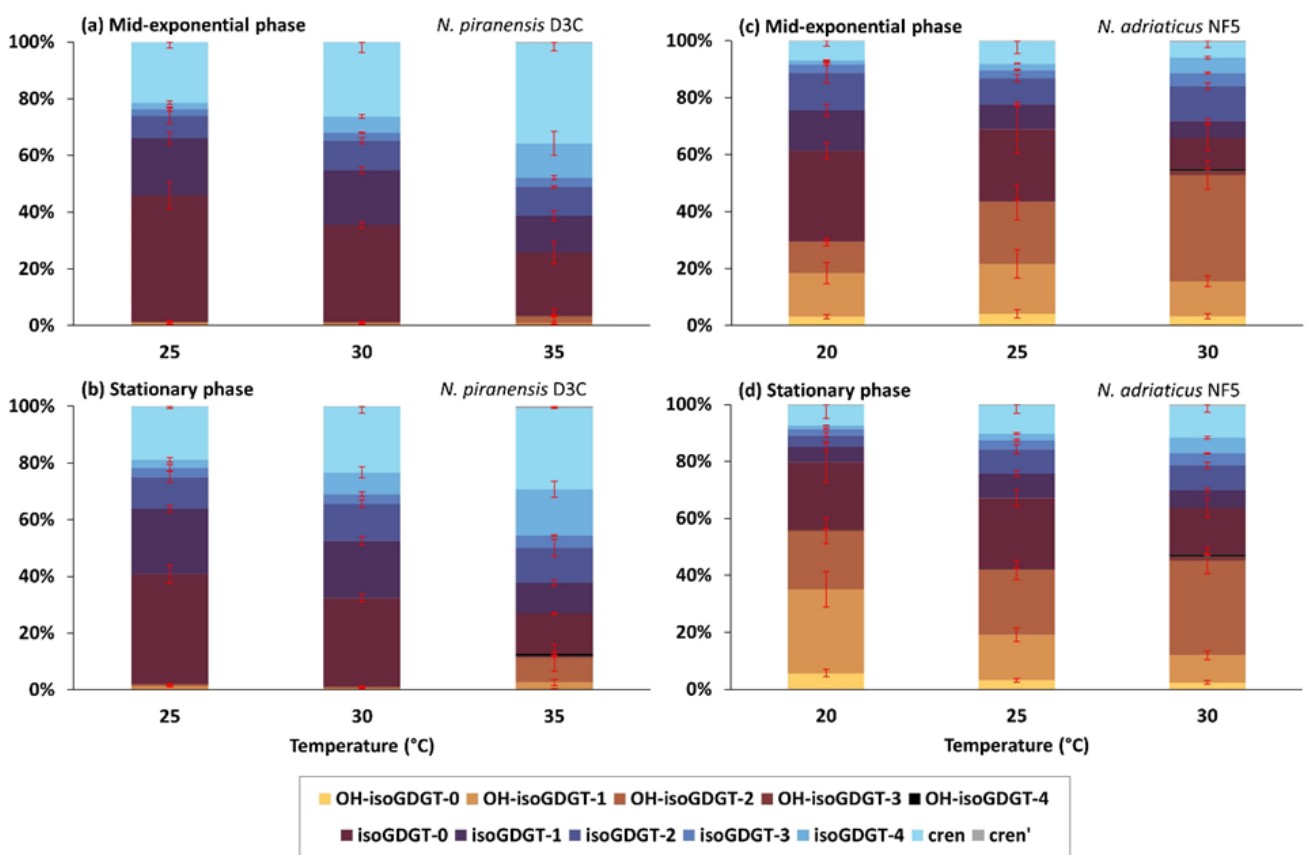


**Figure 1: Relative abundance of core lipids (inferred from IPL analysis), of isoGDGTs and OH-isoGDGTs at different temperatures for (a)** *N. piranensis* **D3C strain harvested at mid-exponential phase, (b)** *N. piranensis* **D3C strain harvested at stationary phase, (c)** *N. adriaticus* **NF5 strain harvested at mid-exponential phase and (d)** *N. adriaticus* **NF5 strain harvested at stationary phase. Error bars denote the standard deviation of the triplicates. Note that since we do not correct for differences in ionization efficiencies for**
**the different headgroups, the actual relative abundance of each head group may be different.**



## 3.2 Impact of growth phase on isoGDGT and OH-isoGDGT distribution

The nitrite concentration of the cultures was used as a proxy to assess their growth phases and the cells were harvested when the nitrite concentration was ~450 µM at mid-exponential phase and ~900 µM at late exponential/ stationary phase (Fig. S3). A cell abundance of $10^7$ and $10^8$ cells ml$^{-1}$ was observed at mid-exponential phase and stationary phase, respectively. The observed variations in isoGDGT and OH-isoGDGT distribution between the mid-exponential and stationary phases were generally small and temperature dependent (Fig. 1). For instance, for *N. piranensis* D3C strain, a slightly higher relative abundance of OH-isoGDGTs during the stationary phase (average of triplicate experiments, 12.6± 7 %) compared to the mid-exponential phase (average of triplicate experiments, 3.5± 3.7 %) is observed at 35 °C (t-test; $p < 0.05$), but not at 25 and 30 °C (Fig. 1a–1b, Table S3). From mid-exponential to stationary phase, isoGDGT-0 and crenarchaeol decrease (from 22.2± 3.9 % to 14.3± 0.3 % and from 35.4± 2.8 % to 28.7± 0.2 % for isoGDGT-0 and crenarchaeol, respectively), while OH-isoGDGT-0 slightly increases (from 0.1± 0.1 % to 0.3± 0.1 %), but only at 35 °C, while a decrease in isoGDGT-0 (from 34.3± 1 % to 31.3± 1.4 %) and an increase in isoGDGT-2 (from 10.4± 1 % to 13.2± 1.3 %) is only observed at 30 °C (t-test; all at a significance of $p < 0.05$). The composition of headgroups does not change from mid-exponential to stationary phase in the *N. piranensis* D3C strain with all three growth temperatures (Fig. 2, Table S4).

The *N. adriaticus* NF5 strain exhibits a higher proportion of OH-isoGDGTs during the stationary phase (average of triplicates 56.3± 12.1 %) compared to the mid-exponential phase (average of triplicates 29.6± 5.4 %) at 20 °C, but not at 25 and 30 °C (Fig. 1, Table S3). At 20 °C, isoGDGT-1 and -2 decreases (from 14.2± 2.1 to 5.5± 1.3 % for isoGDGT-1, and from 13.2± 3.6 to 4± 0.8 % for isoGDGT-2) from mid-exponential to stationary phase (Fig. 1, Table S3). At 30 °C, the relative abundance of isoGDGT-2 decreases (from 12.2± 1.2 % to 8.8± 1.1 %) and crenarchaeol slightly increases (from 5.5± 2 % to 11.2± 2.3 %) from mid-exponential to stationary phase (t-test; all at a significance of $p < 0.05$). However, no trend is observed at 25 °C. In terms of the headgroups, between the mid-exponential and stationary phase across the tested growth temperatures, the fractional abundance of HPH-isoGDGTs do not show any consistent variations, while DH-isoGDGTs generally show a decrease at 20 and 30 °C (from 29.6± 6.2 % to 9± 2.1 % and from 24.6± 2.2 % to 18.2± 1.6 %, respectively) and MH-isoGDGTs show a decrease (from 3.4± 1 % to 2.1± 0.5 %) at 30 °C (t-test; all at a significance of $p < 0.05$). For OH-isoGDGTs, the OH-isoGDGTs with HPH and DH headgroups show an increase from mid-exponential phase to stationary phase (from 0.3± 0.1 % to 0.8± 0.2 % and from 27.9± 5.1 % to 53± 11.3 %, respectively), and correspondingly, MH-OH-isoGDGTs show a decrease (from 1.5± 0.2 to 2.5± 0.6 %), but only at 20 °C (Fig. 2, Table S4).

Our results are in contrast to those observed earlier for *Nitrosopumilus maritimus* SCM1, which showed an increase in MH- and MH-OH-isoGDGTs and a substantial decrease in HPH-isoGDGTs from early growth phase to late stationary phase (Elling et al., 2014). Elling et al. (2014) suggested based on these results that the prevalence of HPH-isoGDGTs indicates growing metabolically active cells. Our results suggest that care has to be taken with this interpretation, as headgroup composition is species-dependent and growth phase differences are not reflected in the abundance of HPH-isoGDGTs relative to MH- and DH-isoGDGTs in our cultures. With respect to core lipids, *N. maritimus* showed a decrease of isoGDGT-0 and crenarchaeol,




and an increase in isoGDGT-2 from exponential phase to stationary phase (Elling et al., 2014). This is similar to some trends
observed in the *N. piranensis* D3C strain, while no such trends are observed for *N. adriaticus* NF5 strain in our study. This is
interesting since *N. maritimus* SCM1 and *N. piranensis* D3C are more closely related to each other phylogenetically than either
of them are to *N. adriaticus* NF5 (Zheng et al., 2024). This suggests that growth phase modulation of IPL and core lipid
composition is not consistent among different species of Thaumarchaeota. Importantly, our results also suggest that growth
phase does not impact OH-isoGDGT abundances and composition in a consistent manner between the mid-exponential and
stationary phases within the range of incubation temperature tested and across different species of the same genus.

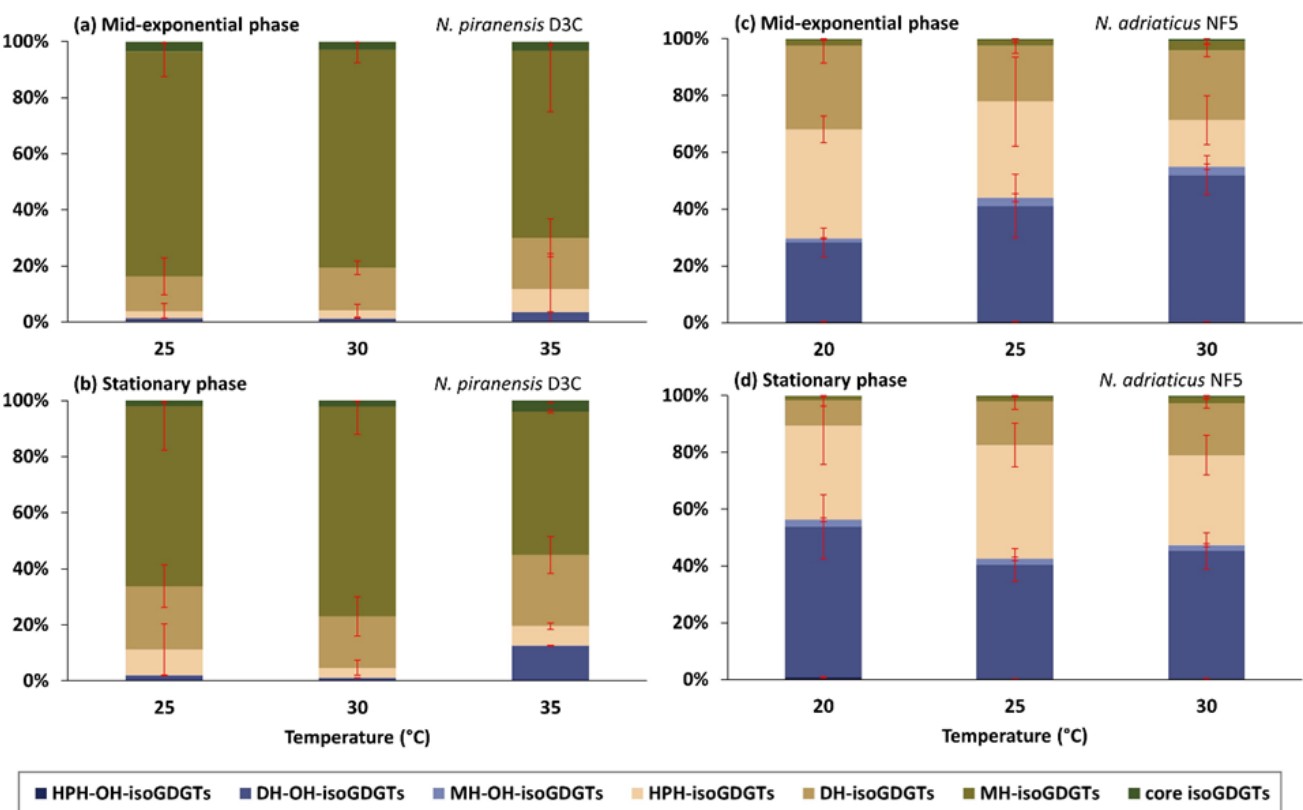

**Figure 2: Relative abundance based on total ion intensity of headgroups of isoGDGTs and OH-isoGDGTs from at different temperatures from (a) *N. piranensis* D3C strain harvested at mid-exponential phase, (b) *N. piranensis* D3C strain harvested at**
**stationary phase, (c,) *N. adriaticus* NF5 strain harvested at mid-exponential phase and (d) *N. adriaticus* D3C strain harvested at stationary phase). HPH, DH and MH denote hexose-phosphohexose, dihexose and monhexose, respectively. Error bars denote the standard deviation of the triplicates Note that since we do not correct for differences in ionization efficiencies for the different headgroups, the actual relative abundance of each head group may be different.**

### 3.3 Impact of growth temperature on isoGDGT and OH-isoGDGT distribution

The two strains were cultivated at different temperatures, at 25, 30 and 35 °C for *N. piranensis* D3C strain, which has an
optimal growth temperature at ~32 °C, and 20, 25 and 30 °C for *N. adriaticus* NF5 strain, which has an optimal growth





temperature at ~30 °C (Bayer et al., 2016). We were unable to grow both strains at temperatures above 35 °C and below 20 °C likely due to their slow growth rate at those temperatures (Bayer et al., 2016).

With respect to isoGDGT core lipids, we observed that for *N. piranensis* D3C strain, isoGDGT-0 and -1 consistently decrease
with increasing temperature while isoGDGT-4 and crenarchaeol slightly increase (Fig. 1, Table S3). For *N. adriaticus* NF5 strain, increasing growth temperature is generally associated with increases in relative abundance of isoGDGT-3, -4 and crenarchaeol isomer, and a decrease in isoGDGT-0 and -1 (Fig. 1, Table S3). In contrast, the relative abundance of crenarchaeol remained low (< 12 %) and did not change with growth temperature. Our results show some similarities but also differences with other culture studies. The *N. maritimus*, NAOA2 and NAOA6 strains showed an increase in the relative abundance of
crenarchaeol and a decrease in isoGDGT-0 and -1 with increasing growth temperature (Elling et al., 2015; Qin et al., 2015), while a decrease in isoGDGT-0 and an increase in crenarchaeol with increasing growth temperature was also observed for *Nitrosopumilus oxyclinae* HCE1 and *Nitrosopumilus cobalaminigenes* HCA1 (Qin et al., 2015), similar to our results for *N. piranensis* D3C strain. However, the thermophilic *Nitrosotenuis uzonensis* exhibited a decrease in isoGDGT-0 to -4 and an increase in crenarchaeol and crenarchaeol isomer from 37 to 46 °C, but showed no change from 46 to 50 °C (Bale et al., 2019).
Looking at the lipid profiles of natural archaeal communities in marine environments, a decrease in isoGDGT-0 and an increase in isoGDGT-1 to -3, crenarchaeol and its isomer is observed with increasing water temperatures (e.g. Kim et al., 2010). Thus, in general, isoGDGT-0 tends to decrease with increasing temperature, while crenarchaeol increases (with the exception of *N. adriaticus* NF5 strain, which has a low abundance of crenarchaeol), while isoGDGT-1 to -4 show variable patterns.

For OH-isoGDGT relative abundances of *N. piranensis* D3C strain, no consistent trends with temperature are observed (Fig.
1, Table S3). Some differences are observed for *N. adriaticus* NF5 strain, e.g. OH-isoGDGT-0 and -1 decrease with increasing growth temperature in stationary phase, OH-isoGDGT-2 and -3 increase with growth temperature in mid-exponential phase and between 25 and 30 °C in stationary phase. Altogether, however, we do not see substantial and consistent changes in the overall abundances of OH-isoGDGTs. If we normalize the OH-isoGDGTs on the total sum of OH-isoGDGTs, we observe that for both *N. piranensis* D3C and *N. adriaticus* NF5 strains, the relative abundance of OH-isoGDGT-0 and -1 decreases (from
10 to 2 % and from 52 to 19 %, respectively for OH-isoGDGT-0 and -1), and OH-isoGDGT-2 increases (from 37 to 70 %) with temperature (Fig. S4). This is similar to the trend observed for OH-isoGDGT core lipids in the marine environment (Lü et al., 2015; Varma et al., 2024).

We calculated the OH-isoGDGT-based proxies, %OH and RI-OH', and regular isoGDGT-based proxies, $TEX_{86}$ and $RI_{isoGDGTs}$, from the core lipid composition of the IPLs for the two strains we investigated (Fig. 3). We observe similar %OH values of
1.5 % at 25 and 30 °C for N. piranensis D3C strain, and a somewhat higher percentage at 35 °C (4 and 13 % in mid-exponential and stationary phases, respectively; Fig. 3). An increase in %OH values with growth temperature is observed in *N. adriaticus* NF5 strain at mid-exponential phase (from 30 to 55 %), while no apparent trend is observed for stationary phase cultures of *N. adriaticus* NF5. Thus, the %OH is not strongly responding to temperature in contrast to the natural marine environment where the relative abundance of OH-isoGDGTs increases (to maximum value of 17 %) with decreasing temperature (e.g. Huguet et
al., 2013; Varma et al., 2024). However, it should be noted that in the marine environment, the relative abundance of OH-





isoGDGTs is quite low at temperatures > 20 °C (< 2 %) and only substantially increases below ~15 °C (Huguet et al., 2013; Varma et al., 2024). These temperatures are much lower than the growth temperatures used in this study (20–35 °C), and thus it may be that only below a certain temperature threshold, the relative abundance of OH-isoGDGTs substantially increases. Unfortunately, we were unable to grow sufficient amount of biomass for both of the thaumarchaeotal species at temperatures

lower than 20 °C because of their extremely low growth rates in those conditions. We also observe that the *N. adriaticus* NF5 strain has an apparent higher relative abundance of OH-isoGDGTs compared to that found in global surface sediments from environments with similar water temperatures (Fig. 3), although, as noted above, quantitative comparisons are difficult due to the fact that we cannot correct for differences in ionization efficiencies of IPLs. More importantly, there are substantial differences in the relative abundances between the two strains, i.e., *N. piranensis* D3C has much lower fractional abundance

of OH-isoGDGTs (1–11 %) than *N. adriaticus* NF5 (29–56 %) (Fig. 1). Calculation of the relative abundances of OH-isoGDGTs in other cultures of Thaumarchaeota (Fig. 4; IPL data from Elling et al., 2015, 2017) also show large variations. Furthermore, for each strain, the OH-isoGDGT abundance at optimal temperatures does not seem to be substantially different from sub-optimal growth temperatures (Fig. S5). Other than *Ca.* Nitrosotalea devanaterra Nd1 having very low %OH value, closely related strains of *Nitrosopumilus* also do not show any particular trend (Fig. S5). This suggests that %OH values may

be impacted by species composition in the natural environment.

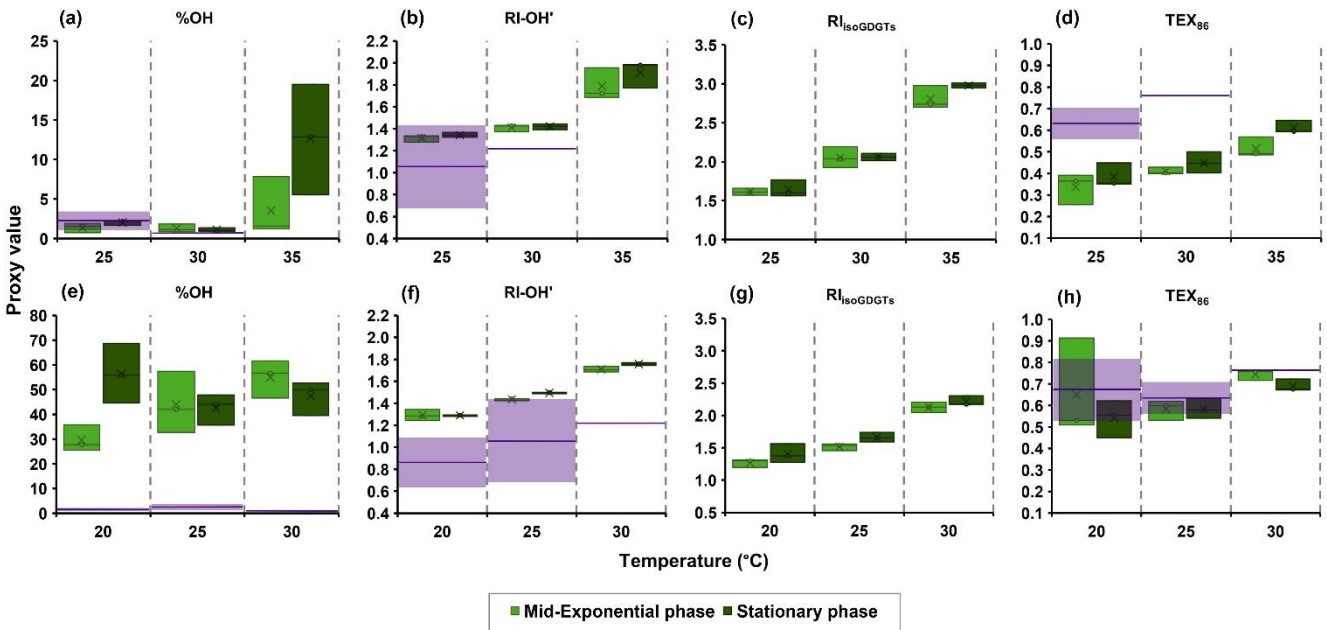

**Figure 3: IsoGDGT and OH-isoGDGT-based proxies with temperature from IPLs of (a–d) *Nitrosopumilus piranensis* D3C strain and (e–h) *Nitrosopumilus adriaticus* NF5 strain, harvested at mid-exponential and stationary phases. Purple lines and shaded areas**

**denote the observed means and one standard deviations, respectively, for each proxy from global marine surface sediments for sea surface temperature (±1 °C) according to compiled datasets based on core lipids from Varma et al. (2024) and Rattanasriampaipong et al. (2022).**



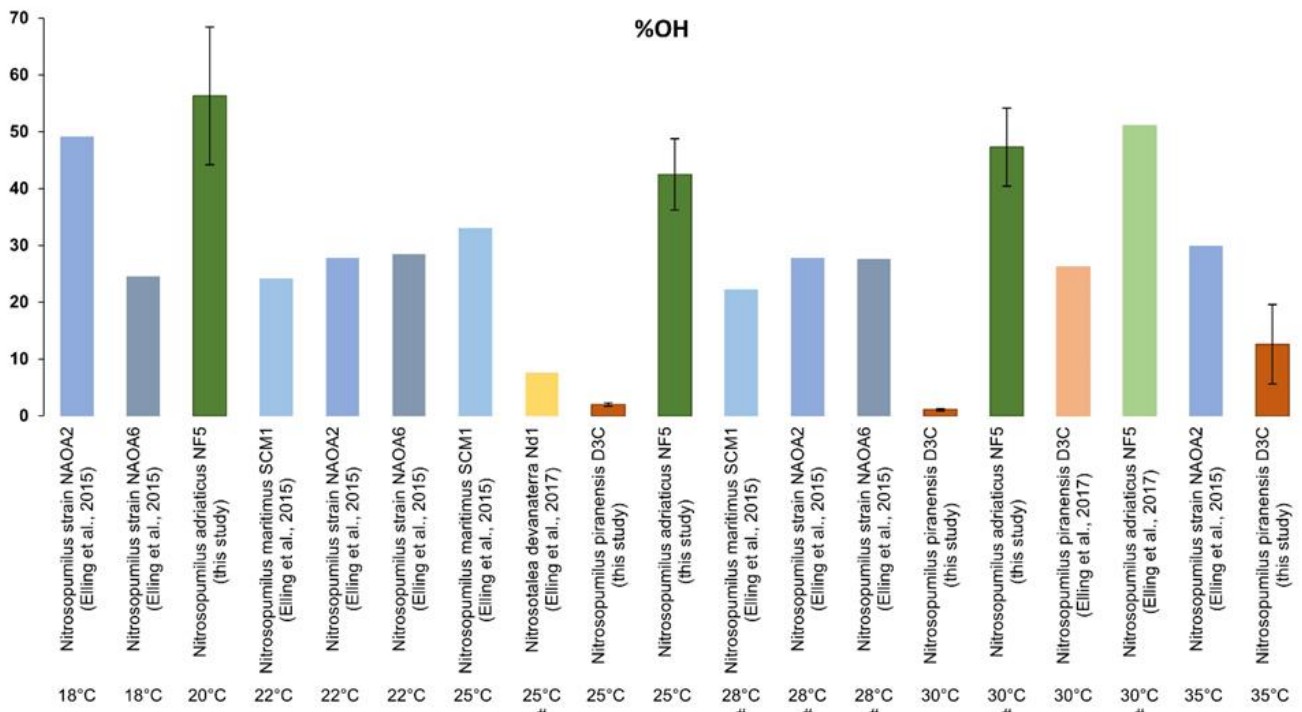

**Figure 4: Relative abundance of OH-isoGDGTs relative to all isoGDGTs plus OH-isoGDGTs inferred from the IPLs, observed in different cultures of Thaumarchaeota grown at different temperatures and harvested at stationary phase. Values from previous studies** (Elling et al., 2017, 2015) **were calculated according to IPL data reported. The optimal temperature of each strain is denoted by #. Error bars denote standard deviation for the triplicates used in this study.**

With respect to RI-OH', we observe a consistent increase with temperature for both *N. piranensis* D3C and *N. adriaticus* NF5 strains, with values slightly higher than RI-OH' values observed for core lipids in marine surface sediments from a similar sea surface temperature regime (Fig. 3). Interestingly, since the sedimentary OH-isoGDGT signal, similar to regular isoGDGTs, are more likely to be coming from subsurface waters with lower temperatures (cf. Varma et al., 2024), this would align the environmental RI-OH' values with the observed RI-OH' values in the cultures. For the regular isoGDGTs, the $RI_{isoGDGTs}$ increases with temperature for both strains, while for $TEX_{86}$, *N. piranensis* D3C strain shows an increase with increasing temperature, but *N. adriaticus* NF5 strain showed only an increase in $TEX_{86}$ from 25 to 30 °C (Fig. 3). This agrees with previous observations (Bale et al., 2019; Elling et al., 2015; Qin et al., 2015) where $RI_{isoGDGTs}$ shows a better response to temperature than $TEX_{86}$ for different cultures of Thaumarchaeota. The $TEX_{86}^{OH}$ also shows a similar trend with temperature as $TEX_{86}$ for *N. piranensis* D3C strain due to the low abundance of OH-isoGDGTs in this strain (Fig. S6). However, in *N. adriaticus* NF5 strain, the $TEX_{86}^{OH}$ shows an increasing trend with temperature in stationary phase unlike $TEX_{86}$, although this is not observed for mid-exponential phase. Thus, the ring indices of both isoGDGTs and OH-isoGDGTs consistently increase

with growth temperature, while the relative abundance of OH-isoGDGTs seems independent of growth temperature, at least for the temperature range and species used in this study.

**3.4 Comparison of IPLs with different headgroups**

The headgroup compositions do not exhibit a consistent change with temperature for both *N. piranensis* D3C and *N. adriaticus*
NF5 strains (Figs. 2, S1, S2, Table S4). This is similar to what was reported for *N. maritimus*, but NAOA6 showed a considerable decrease in MH-isoGDGTs while NAOA2 showed increase in HPH-isoGDGTs with temperature (Elling et al., 2015). Bale et al. (2019) observed no substantial change in headgroup composition in *N. uzonensis* between 37 and 46 °C, while an increase in MH and DH headgroups of isoGDGTs and a decrease in HPH-isoGDGTs were observed between 46 and 50 °C. This suggests that headgroup composition may not be influenced by changes in temperature in a consistent manner but
mainly differs between different strains.

Interestingly, we do observe large differences in distribution of core lipids between the different headgroups (Figs. S1, S2). Indeed, different proxy values and trends are observed when they are calculated for each headgroup (Figs. 5, S7). For example, $\%OH$, $RI\text{-}OH'$, $TEX_{86}^{OH}$ and $TEX_{86}$ for the DH headgroup have substantially higher values compared to those of other headgroups. This observation is not unusual, for example, the ring index of isoGDGTs and $TEX_{86}$ values have been shown to
vary strongly between IPLs with different headgroups in culture studies as well as in water column and sediments (Elling et al., 2014; Lengger et al., 2012; Schouten et al., 2012; Basse et al., 2014b). Comparison of *N. piranensis* D3C and *N. adriaticus* NF5 strains shows that OH-isoGDGTs mainly occur with DH headgroup and only to a minor extend with MH and HPH (Fig. 2). This agrees with the results of the same strains from Elling et al. (2017) who also observed a predominance of OH-isoGDGTs in IPLs with a DH headgroup.

With respect to $TEX_{86}^{OH}$, $TEX_{86}$ and $RI_{isoGDGTs}$, all of them increase with increasing temperature for all three headgroups for *N. piranensis* D3C and *N. adriaticus* NF5 strains, and only $TEX_{86}$ and $TEX_{86}^{OH}$ based on core lipids show no increase (Figs. 5, S7). In contrast, $RI_{isoGDGTs}$ for *N. maritimus*, NAOA6 and NAOA2 show variable trends in the response of different IPLs to increasing growth temperature (Elling et al., 2015). For OH-isoGDGT-based proxies, some trends in $\%OH$ and $RI\text{-}OH'$ are more prominent for specific IPLs. For example, a substantially higher $\%OH$ is observed for IPLs with a DH headgroup in both
strains, and for *N. adriaticus* NF5 strain, IPLs with MH and DH headgroups show a slight decrease in $\%OH$ with increasing growth temperature (Fig. 5). For $RI\text{-}OH'$, a slight increase with temperature is observed for IPLs with DH and MH groups in *N. piranensis* D3C strain, and for *N. adriaticus* NF5 strain this trend is apparent for all IPL headgroups (Figs. 3, 5).

The fact that OH-isoGDGT-derived proxies show such strong differences between IPLs with different headgroups may become relevant when IPLs are transformed to core lipids upon cell death and settling to the sediment floor in the natural environment.
IPLs are known to have substantially different degradation rates, with HPH IPLs thought to degrade much faster than IPLs with MH or DH headgroups (Schouten et al., 2012; Xie et al., 2014). If so, then this could substantially alter the amount and possibly even distribution of OH-isoGDGTs upon degradation, with OH-isoGDGTs initially being present at lower abundances



as fossil lipids compared to how they occurred as IPLs in the living cell as they are mainly tied to the DH headgroup, which degrades much slower than IPLs with the HPH headgroup, containing hardly any OH-isoGDGTs (Figs. 2, 5a, 5e).


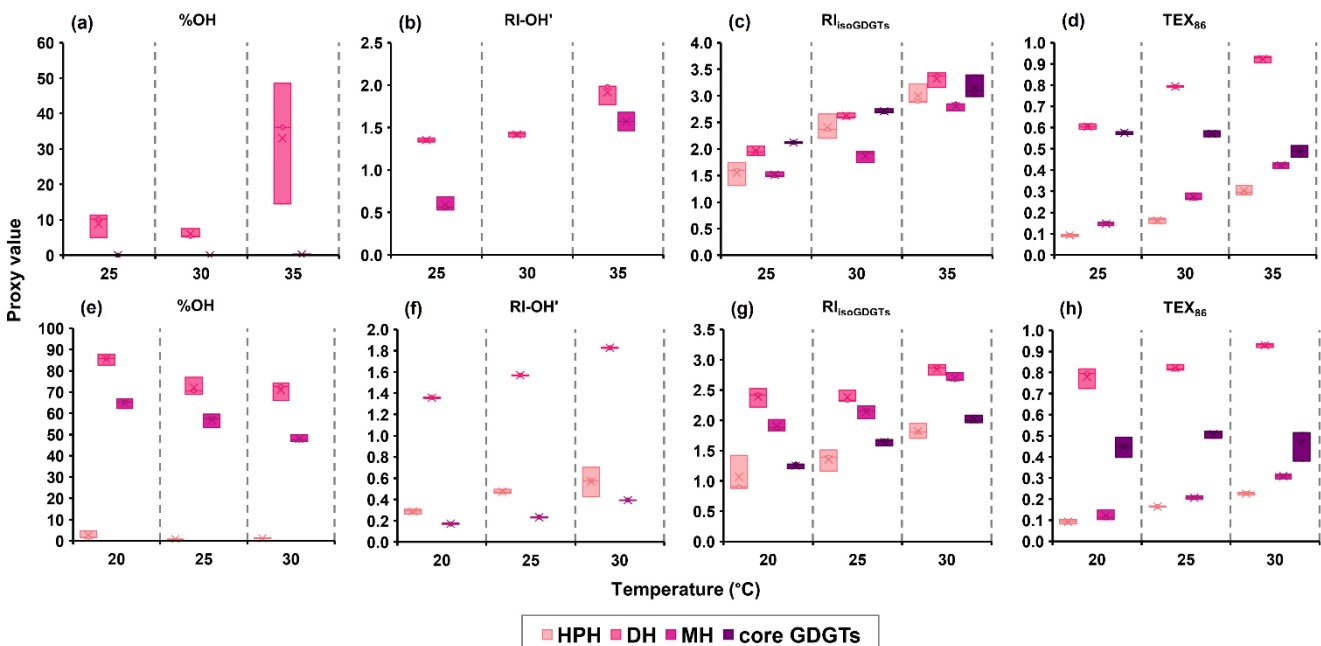

**Figure 5: Correlations of isoGDGT and OH-isoGDGT-based proxies with temperature for different headgroups of IPLs from (a–d)** *Nitrosopumilus piranensis* **D3C strain and (e–h)** *Nitrosopumilus adriaticus* **NF5 strain, harvested at stationary phase.**

## 4 Conclusion

This study shows how the lipid composition, specifically OH-isoGDGTs and the proxies derived from them, are affected by growth temperature and growth phase in two strains of Thaumarchaeota. No systematic trends are observed in the core lipid and headgroup composition between the mid-exponential and stationary phases for both *N. piranensis* D3C and *N. adriaticus* NF5 strains. Growth temperatures do cause some changes in lipid composition, mainly in the degree of cyclization of isoGDGTs and OH-isoGDGTs. Indeed, $RI - OH'$ increases with increasing temperature in both cultures, similar to the marine

environment, in contrast to %OH which remains relatively constant. The *N. adriaticus* NF5 strain shows a substantially higher abundance of OH-isoGDGTs compared to *N. piranensis* D3C, suggesting species composition impacts %OH. OH-isoGDGT-based proxies calculated from intact polar lipids with different headgroups also show considerable differences between the headgroups and variations in their response to temperature in both strains, similar to regular isoGDGT-based proxies. This suggests that species composition and variations in IPLs and their degradation rates may have an impact on %OH in the natural

environment. Further studies investigating the OH-isoGDGTs in archaeal cultures grown at temperatures < 15 °C, e.g. from

species isolated from cold/ polar regions, are needed to provide insights into the role of OH-isoGDGTs in archaeal membrane adaptation.

## Data availability

All data related to this article is available online through the DOI: https://dataportal.nioz.nl/doi/10.25850/nioz/7b.b.kh. The
data is also available in the supplementary materials.

## Author contributions

SS, LV and DV designed the experiments. DV conducted lab experiments which included culturing work, lipid extraction and analysis. NJB assisted with lipid data analysis from UHPLC. DV the performed the data analysis and wrote the manuscript together with all co-authors. SS, LV, PO and GJR supervised the project, provided critical feedback and contributed to shaping
the research and manuscript.

## Competing interests

The authors declare that they have no conflict of interest.

## Acknowledgements

We thank Barabara Bayer for providing the cultures used in this study and Monique Verweij and Denise Dorhout for analytical
support. This work was carried out under the program of the Netherlands Earth System Science Centre (NESSC), financially supported by the Ministry of Education, Culture and Science (OCW) through grant 024.002.001 to SS. This project has received funding from the European Union's Horizon 2020 research and innovation program under the Marie Skłodowska-Curie, grant agreement No 847504.

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
