# Peer review of "Controls on the composition of hydroxylated isoGDGTs in cultivated ammonia oxidizing Thaumarchaeota"

_EGUsphere, 2024_

## Author Comment (AC1)

*We want to thank reviewer #1 for their comments and providing constructive feedback on our manuscript. Below we address all the comments with our replies in italics.*

Reviewer #1: This manuscript uses cultures of two Thaumarchaeotal strains to investigate growth temperature and phase on OH-GDGT distributions and OH-GDGT-based proxies. This is an important piece of work in the context of the emergence of OH-GDGTs as a GDGT-based temperature indicator, especially in cold water settings. The paper is clear and well written with nicely presented figures, and thoroughly reported results. I mostly only have a few very minor suggested comments/changes, but did feel like the manuscript was missing an 'implications' or 'summary' section to tie the results together and put them into a wider context. Could the authors comment on what they see as the key implications of interspecies variability in OH-GDGTS with changes in growth temperature for OH-GDGT-based temperature proxies?

*We thank the reviewer for the positive feedback on our manuscript and we agree that including an Implications/ Summary section would be a valuable addition. Therefore, we will add this in the revised manuscript.*

Specific comments below:

Line 23. Add 'the' before natural.

*We will change this in the revised manuscript.*

Line 32. Add a comma between 'moieties' and 'and'.

*We will change this in the revised manuscript.*

Line 43. Add 'the' before Black Sea.

*We will change this in the revised manuscript.*

Line 141. Correct format for tex86oh.

*We will change this in the revised manuscript.*

Figure 3: Is there a reason purple shaded boxes haven't been added to c) and g)? It would be good to see how these results compare to global core top data sets for RI isoGDGTs.

*The reason why mean and standard deviations in purple shaded boxes were not presented for $RI_{isoGDGTs}$ is because in this study we calculated the ring index of isoGDGTs based on Equation 5 according to (Pearson et al., 2004) where isoGDGT-4 was included. However, most studies in literature have not reported the isoGDGT-4 data and therefore, mean and standard deviation of $RI_{isoGDGTs}$ could not be calculated for global marine surface sediments to present here.*

Line 347: Extent rather than extend.

*We will make this modification.*

Discussion: It feels like an implications section is missing here, that sums up the results and discussion and distills out the key points and implications of these findings for future OH-GDGT- based research. I recommend adding a paragraph to this effect before the conclusions.

*We will add this as mentioned in the general comments above.*

**References**

*Pearson A., Huang Z., Ingalls A. E., Romanek C. S., Wiegel J., Freeman K. H., Smittenberg R. H. and Zhang*

*C. L. (2004) Nonmarine crenarchaeol in Nevada hot springs. Appl. Environ. Microbiol. **70**, 5229–5237.*

---

## Author Comment (AC2)

*We want to thank reviewer #2 for their comments and providing constructive feedback on our manuscript. Below we address all the comments with our replies in italics.*

Reviewer #2: This manuscript systematically investigates the response of hydroxylated GDGTs (OH-GDGTs) in ammonia-oxidizing archaea to temperature variations and different growth phases, revealing several significant findings. The manuscript is well-written, the data are presented clearly, and the conclusions are substantial and scientifically meaningful. However, I have two main concerns and a few suggestions for further research:

1. The study uses the Bligh-Dyer method for lipid extraction, and it is mentioned that this method may lead to partial dehydration of hydroxylated isoGDGTs, potentially underestimating the abundance of OH-isoGDGTs. This could partly explain the discrepancies observed between experimental data and environmental samples. I recommend that the authors discuss the potential impact of this extraction method on the results more thoroughly and consider whether alternative extraction methods might more accurately reflect the GDGT composition in pure cultures. I would recommend considering the use of cellular acid hydrolysis in future studies to investigate the response of core lipids systematically in the future. This approach may provide a more comprehensive understanding of how core lipids behave under different conditions and could help to address some of the limitations observed with the current extraction methods.

*We would like to clarify that we used modified Bligh-Dyer method for lipid analysis and analyzed intact polar lipids (IPLs) to actually avoid dehydration of hydroxy GDGTs as this method does not involve any strong acids. In contrast, core lipid analysis after acid hydrolysis of IPLs, used in other culture studies/ environmental studies, can lead to partial dehydration of these OH-isoGDGTs, which would result in underestimation of OH-isoGDGT abundance. Therefore, we recommend using Bligh-Dyer method when analyzing OH-isoGDGTs from cultures instead of using acid hydrolysis. We will rephrase the sentence to clarify this better in the manuscript to avoid this apparent confusion.*

2. Figure 1 shows that the N. piranensis D3C strain exhibits some differences at higher temperatures, particularly at 35°C, where the relative abundance of OH-isoGDGTs differs significantly between growth phases. However, the discussion of this aspect in the manuscript is somewhat brief. I suggest the authors provide a more detailed explanation of the mechanisms by which high temperatures might influence these differences, especially when compared with environmental observations.

*It is true that N. piranensis D3C strain exhibits a higher relative abundance of OH-isoGDGTs at stationary phase compared to mid-exponential phase at 35 °C, but not at 25 and 30 °C. However, we would like to highlight that this is not a specific case for N. piranensis D3C strain alone, but also observed in N. adriaticus NF5, where a higher relative abundance of OH-isoGDGTs was observed in stationary phase compared to mid-exponential phase at 20 °C, but not at 25 and 30 °C. Thus, there does not seem have a particular trend that differentiates between growth phases which are more apparent at higher temperatures. Therefore, we refrain from speculating on a general mechanism to explain this isolated observation for N. piranensis. Furthermore, we don't have observations of growth phases in the natural environment and hence it is very difficult to compare these isolated culture observations to the natural environment.*

3. Given that the study found inconsistencies between the behavior of OH-GDGTs in the experimental setting and in environmental samples, particularly regarding temperature's effect on cyclization but limited impact on OH%, future research could focus on the performance of different strains at lower temperatures (<15°C), especially those isolated from cold/polar regions. This could provide valuable insights into the physiological role of OH-GDGTs in archaeal membrane adaptation and further refine the proxies based on them.

*We fully agree that future research should focus on studying OH-isoGDGTs in archaeal cultures grown at temperatures < 15 °C and in cultures from polar/sub-polar regions. We have mentioned this in the Conclusions section.*

I believe that this manuscript makes a significant contribution to our understanding of OH-GDGT proxies and is worthy of publication.

*We appreciate the reviewer's positive comments on our manuscript.*